# Improvement of growth, lipid metabolism and innate immune response in Pacific white shrimp (*Penaeus vannamei*) post-larvae through enrichment of live feeds with *Schizochytrium* sp., taurine and inosine monophosphate

Suhyeok Kim[1], Sanghyun Song[1], Yeonji Lee[1], Yein Lee[1], Young-Mi Lee🔵[2], Kook-Jin Cho[2], Kyeong-Jun Lee🔵[1,3]*

1 Department of Marine Life Sciences, Jeju National University, Jeju, South Korea, 2 CJ Cheiljedang Co., Seoul, South Korea, 3 Marine Life Research Institute, Jeju National University, Jeju, South Korea

* kjlee@jejunu.ac.kr

## Abstract

Nutritional deficiencies during the early developmental stages of *Penaeus vannamei* often lead to poor growth, weak immunity and high mortality. To address these issues, this study investigated the effects of live feeds enriched with *Schizochytrium* sp. (SCH), taurine and inosine monophosphate (IMP) on the growth, lipid metabolism and immune response of shrimp post-larvae. Two feeding trials were conducted using rotifers (Trial 1) and *Artemia* (Trial 2) as live feeds. In trial 1, post-larvae (PL1–2) were fed unenriched rotifers (RN) or rotifers enriched with 0.5% SCH (RS), RS + 0.1% taurine (RS + T), RS + 0.1% IMP (RS + I) and 0.5% *Chlorella* powder (RCL) for 9 days. In trial 2, post-larvae (PL12–14) were fed unenriched *Artemia* (AN) or *Artemia* enriched with 0.5% SCH (AS), AS + 0.1% taurine (AS+T), AS + 0.1% IMP (AS+I) and 0.5% *Chlorella* powder (ACL) for 12 days. In trial 1, growth was significantly higher in RS, RS + T and RS + I groups compared to RN and RCL groups. Survival was significantly higher in RS, RS + T and RS + I groups than in RN group. In trial 2, growth performance was significantly higher in all SCH-enriched groups compared to AN and ACL groups whereas, survival did not differ significantly among the experimental groups. In both trials, whole-body docosahexaenoic acid levels increased markedly in all SCH-supplemented treatments, and the expression of lipid metabolism-related genes (*FAS*, *FABP*, *FATP* and *CPT1*) and immune-related genes (*lysozyme*, *crustin*, *LGBP*, *Pen-3a* and *proPO*) was significantly upregulated. These results demonstrate that enrichment of live feeds with SCH effectively enhances growth, lipid utilization and immune function in *P. vannamei* post-larvae. Furthermore, combined supplementation with taurine and IMP produced synergistic immunostimulatory effects. Overall, SCH-based enrichment can serve as a promising functional nutritional fortifier for shrimp hatchery diets.

**Data availability statement:** All relevant data are within the paper.

**Funding:** The funders contributed only by producing and supplying the Schizochytrium powder (CJ Cheiljedang Co., Ltd.) and by providing experimental tanks and facilities (National Research Foundation of Korea). The funders had no role in study design, data collection and analysis, decision to publish, or preparation of the manuscript.

**Competing interests:** The authors declare that they have no known competing financial interests or personal relationships that could have appeared to influence the work reported in this paper. This does not alter our adherence to PLOS ONE policies on sharing data and materials.

## Introduction

Pacific white shrimp (*Penaeus vannamei*) is the most widely farmed shrimp species globally, with production increasing steadily in recent years [1]. In shrimp hatcheries, the quality of live feeds is crucial for normal development and optimal growth of the shrimp post-larvae (PL) [2]. Rotifer (*Brachionus plicatilis*) and *Artemia* are the most commonly used live feeds due to their high digestibility and ability to stimulate feeding responses and minimal impact on the water quality [3,4]. However, they lack essential nutrients such as essential fatty acids, amino acids and vitamins [5]. When the rotifers and *Artemia* are used as larval feeds, their nutritional enrichment is usually needed to ensure that the larvae can receive the required essential nutrients [3].

Microalgae are rich in essential fatty acids, particularly docosahexaenoic acid (DHA, C22:6n3) and eicosapentaenoic acid (EPA, C20:5n3), and have been widely recognized as effective nutritional sources for enriching live feeds to improve growth, lipid metabolism and immunity in aquatic animal larvae [5]. They are easy to cultivate and can be produced in large quantities, with high protein (30–50%) and lipid (10–20%) levels as well as an abundance of vitamins and carotenoids, making them highly valuable for the larval culture [6]. *Schizochytrium* sp. (SCH), a type of microalgae, has gained considerable attention as a potential substitute for fish oil and a functional additive in diets for Atlantic salmon (*Salmo salar*) and Pacific white shrimp due to its high level of DHA [7,8]. SCH is also rich in astaxanthin, niacin and pyridoxine, which play important roles in antioxidant capacity, immunity and disease resistance of aquatic animals [5]. Previous studies have demonstrated that enrichment of *Artemia* with SCH improved growth, stress resistance, and survival in finfish species such as turbot (*Scophthalmus maximus*) and silver pompano (*Trachinotus blochii*) [9,10]. However, limited studies have examined its application in Pacific white shrimp. Another microalga commonly used for enrichment, *Chlorella* sp., is inexpensive and easy to culture [11], but contains little DHA, which is essential for shrimp larval growth and survival [12]. Although DHA can be synthesized from dietary α-linolenic acid (C18:3n3) through sequential reactions catalyzed by Δ4/Δ6 desaturases and elongases of very long-chain fatty acids 5, leading to the formation of EPA and docosapentaenoic acid (DPA, C22:5n3), or alternatively via the Sprecher pathway from EPA through elongation, Δ6 desaturation and β-oxidation, the endogenous production is insufficient to fully support optimal growth [13,14]. Thus, a supplemental dietary DHA intake is crucial to optimize and/or maximize the growth and survival of the shrimp PL [14]. To address this concern, two consecutive studies were conducted to examine the potential application of SCH as a DHA source through the nutrition enrichment of the rotifers (the first experiment, EXP-1) and *Artemia* (the second experiment, EXP-2).

Taurine plays an important role in osmoregulation, bile salt formation and modulating cellular processes like oxidative stress and immune responses [15]. In aquatic animals, taurine has been shown to improve growth performance and stress resistance by stabilizing cellular membranes and promoting the immune functions [16]. Inosine monophosphate (IMP) is a purine nucleotide involved in cellular energy metabolism, serving as a precursor for adenosine triphosphate synthesis [17].

IMP plays a role in the enhancement of growth, stress resistance and survival by supporting mitochondrial function and improving cellular energy reserves [18]. These two supplements were expected to improve not only the nutritional quality of the live feeds but also the shrimp immunity and lipid metabolism. To date, no study has evaluated the combined effects of SCH, taurine and IMP in live feed enrichment for crustaceans. Therefore, this study aimed to investigate whether enrichment of rotifers and *Artemia* with SCH, taurine and IMP could improve growth, lipid metabolism, immunity and survival in Pacific white shrimp PL. This research provides novel insight into functional live feed enrichment strategies to enhance shrimp larval performance and health.

## Materials and methods

### *Schizochytrium* powder

SCH was provided by CJ CheilJedang (Seoul, South Korea). The SCH strain used in this study was isolated from the South Sea of South Korea and cultured in GYPS medium comprising 20 g/L D-glucose, 2 g/L yeast extract, 6 g/L peptone, 17 g/L sucrose, 12 g/L sea salt and 15 g/L bacto-agar (KR 20240026429A) [19]. The strain was characterized and deposited at the Korea Collection for Type Cultures (KCTC) under the accession number KCTC15006 BP. For scale-up production, the microalgae were cultivated in a 3,000 L fermenter with glucose supplementation. The fermentation broth was subsequently processed in a granulator to produce dried biomass. The SCH powder is a fine, yellow–brown spray-dried microalgal powder (particle size: 50–200 μm) with a moisture content of <5%. The proximate composition (crude lipid, crude protein and ash) and fatty acid profiles are provided in Table 1.

### Rotifers culture and enrichment

Rotifers were stocked in a tank (50 L) at a density of 5,000–10,000 individuals/mL and fed with *Chlorella vulgaris* powder (Ecofactory Co., Ltd., Seoul, South Korea). The environmental conditions for rotifers culture were

**Table 1. Proximate composition and fatty acid profiles of enrichment sources.**

|  | *Chlorella* powder | SCH | SCH+T | SCH+I |
|---|---|---|---|---|
| *Proximate composition (%, dry matter)* | | | | |
| Crude protein | 63.0 | 70.5 | 70.6 | 70.5 |
| Crude lipid | 7.18 | 20.1 | 20.1 | 20.1 |
| Crude ash | 7.07 | 8.37 | 8.14 | 8.47 |
| *Fatty acid (%, lipid)* | | | | |
| C14 | ND | 2.28 | 2.27 | 2.28 |
| C16 | 20.1 | 52.9 | 51.9 | 53.1 |
| C17:1 | 20.5 | ND | ND | ND |
| C18 | ND | 2.29 | 2.30 | 2.28 |
| C18:1n9c | 5.28 | ND | ND | ND |
| C18:2n6c | 50.3 | ND | ND | ND |
| C18:3n3 | 3.84 | ND | ND | ND |
| C22:6n3 | ND | 42.6 | 43.5 | 42.3 |
| ∑ Saturated fatty acids[1] | 20.1 | 57.5 | 56.5 | 57.7 |
| ∑ Monounsaturated fatty acids[2] | 25.8 | ND | ND | ND |
| ∑ Polyunsaturated fatty acids[3] | 54.1 | 42.6 | 43.5 | 42.3 |

Abbreviations: SCH, *Schizochytrium*; SCH+T, *Schizochytrium* added with 0.1% taurine; SCH+I, *Schizochytrium* added with 0.1% inosine monophosphate; ND, not detected.

maintained as water temperature 25.5 ± 0.1 °C, dissolved oxygen (DO) 6.6 ± 0.6 mg/L, salinity 25.0 ± 0.2 ppt and pH 7.5 ± 0.5. Rotifers were starved for 8 h prior to enrichment to ensure that any *Chlorella* previously ingested was cleared from their guts. The rotifers were then removed from the culture tanks, rinsed with seawater and transferred to small culture containers (2 L) with aeration. During the enrichment period, water quality parameters were maintained as per the previous culture conditions. Enrichment was carried out for 2 h with the addition of materials for each experimental group. Five experimental rotifer groups consisted of non-enriched group (RN) and enriched groups with 0.5% SCH (RS), RS + 0.1% taurine (RS+T), RS + 0.1% IMP (RS+I) and 0.5% *Chlorella* powder (RCL). Each experimental group was conducted in quadruplicate. The enriched rotifers were transferred to a 55 μm mesh filter and carefully washed with seawater. The rotifers were examined under an optical microscope (DM750, Leica Microsystems) fitted with a digital camera (ICC50E, Leica Microsystems, Heerbrugg, Switzerland) to verify the enrichment (Fig 1). After shrimp feeding, the remaining rotifers were stored at −82 °C for the analysis of proximate composition and fatty acid profiles.

### *Artemia* culture and enrichment

*Artemia* cysts (Sep-art *Artemia*, INVE, Salt Lake City, UT, USA) were hatched using a commercial incubator (2 L), cultured and enriched [20]. The hatching conditions were density 1.0 g/L, water temperature 30.1 ± 0.1 °C, salinity 30.5 ± 0.1 ppt and pH 7.0 ± 0.5 for 24 h. After hatching, a magnetic bar was used to remove the shells. *Artemia* were allowed to consume the yolk for 22 h. Enrichment was then performed for 2 h under each enrichment condition. Five experimental *Artemia* groups consisted of non-enriched group (AN) and enriched groups with 0.5% SCH (AS), AS + 0.1% taurine (AS+T), AS + 0.1% IMP (AS+I) and 0.5% *Chlorella* powder (ACL). Each experimental group was conducted in quadruplicate. The enriched *Artemia* were collected using a 0.1 mm sieve and then rinsed with freshwater to remove debris. The *Artemia* were examined under an optical microscope (DM750, Leica Microsystems) fitted with a digital camera (ICC50E, Leica Microsystems, Heerbrugg, Switzerland) to verify the enrichment (Fig 2). The remaining *Artemia* after feeding to the shrimp were stored at −82 °C for analysis of proximate composition and fatty acid.

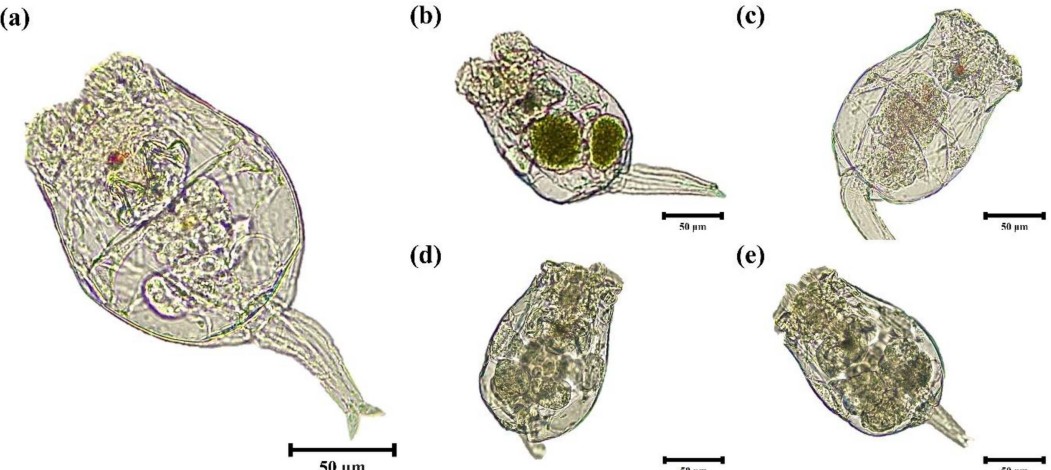

**Fig 1. Microscopic visualization of the rotifers *Brachionus rotundiformis* after 2 h enrichment with each treatment.** (a) non-enriched rotifers, (b) enriched rotifers by *chlorella* powder, (c) enriched rotifers by *Schizochytrium*, (d) enriched rotifers by *Schizochytrium* with 0.1% taurine, (e) enriched rotifers by *Schizochytrium* with 0.1% inosine monophosphate. Scale bar: 50 μm; Original magnification × 100.

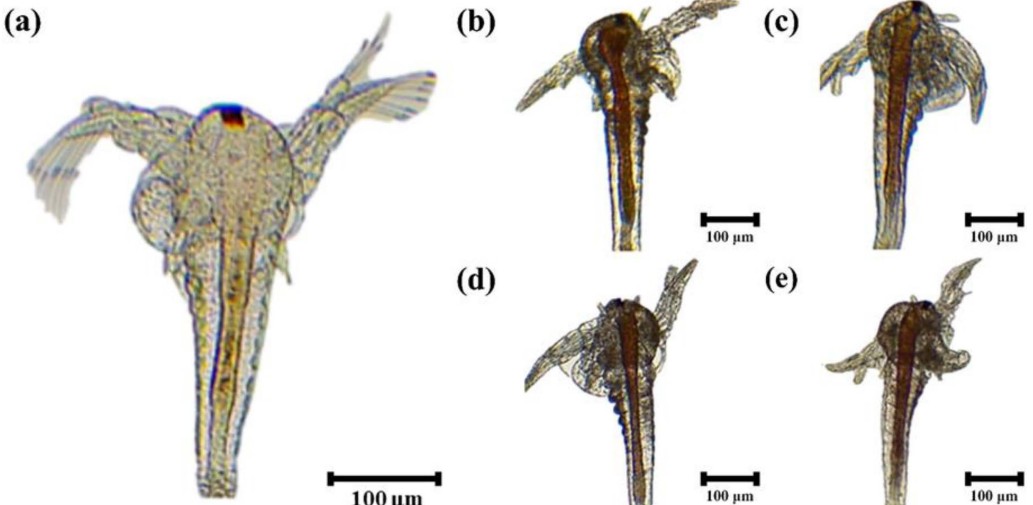

**Fig 2. Microscopic visualization of the *Artemia* after 2 h enrichment with each treatment.** (a) non-enriched *Artemia*, (b) enriched *Artemia* by *chlorella* powder, (c) enriched *Artemia* by *Schizochytrium*, (d) enriched *Artemia* by *Schizochytrium* with 0.1% taurine, (e) enriched *Artemia* by *Schizochytrium* with 0.1% inosine monophosphate. Scale bar: 100 μm; Original magnification × 40.

## Feeding trial

Pacific white shrimp were obtained from a shrimp hatchery (Muan, South Korea). The feeding trials were conducted at the Kidang Marine Science Institute, Jeju National University, Jeju, South Korea. In the EXP-1, a total of 4,000 shrimp at the PL1−2 stage were distributed at a density of 200 shrimp per tank (acrylic, 96 L), with four replicates per experimental group. The initial body length of the shrimp was $0.67 \pm 0.03$ mm. Aeration was set up for each tank to provide DO and the photoperiod was maintained at 12 h dark:12 h light using fluorescent lamps. DO, temperature, pH, salinity and ammonia were measured three times daily using a ProQuatro Multiparameter Meter (YSI, Yellow Springs, OH, USA). During the EXP-1, water quality parameters were maintained as DO $6.43 \pm 0.30$ mg/L, water temperature $30.01 \pm 0.98$ °C, pH $7.2 \pm 0.7$ and ammonia $0.06 \pm 0.01$ mg/L. At the beginning of EXP-1, the rotifer feeding amount was 80 rotifers per shrimp, which was increased by 20% daily. The shrimp were fed the experimental rotifers three times a day (8:30, 14:30 and 20:30 h) for 9 days.

In the EXP-2, a total of 2,000 shrimp at the PL12−14 stage were distributed at a density of 100 shrimp per tank (acrylic, 96 L), with four replicates per experimental group. The initial body weight of the shrimp was $3.67 \pm 0.23$ mg. The experimental environment was set up similarly to EXP-1. DO, temperature, pH, salinity and ammonia were measured three times daily using a ProQuatro Multiparameter Meter (YSI, Yellow Springs, OH, USA). During the EXP-2, water quality parameters were maintained as DO $6.87 \pm 0.21$ mg/L, water temperature $29.81 \pm 0.89$ °C, pH $7.0 \pm 0.4$ and ammonia $0.08 \pm 0.02$ mg/L. At the beginning of EXP-2, the *Artemia* feeding amount was 80 *Artemia* per shrimp, which was increased by 10% daily. The shrimp were fed the experimental *Artemia* three times a day (8:30, 14:30 and 20:30 h) for 12 days.

## Proximate composition and fatty acid profile

The proximate composition of SCH, rotifers and *Artemia* was analyzed based on the AOAC [21] method. Moisture was measured by the amount remaining after drying at 125 °C for 3 h. Crude ash was measured by the amount remaining after roasting at 550 °C for 4 h. Crude protein was determined using a Kjeldahl nitrogen digester (SpeedDigester K-439, Buchi,

Gwangmyeong, South Korea). Crude lipid was analyzed based on the Folch et al. [22] method. The fatty acid profile of enrichment sources, enriched rotifers, enriched *Artemia* and whole-body of shrimp was analyzed by the gas chromatography method [23] using an 800GC (Agilent, CA, USA).

### Growth performance and survival

At the end of the feeding trials, all shrimp were individually measured for length, weight and number. In EXP-1, length gain (LG) was calculated based on the final body length (FBL) of the shrimp. In EXP-2, weight gain (WG) was calculated based on the final body weight (FBW) of the shrimp. Survival was calculated based on the final number of shrimp in both experiments.

### Hepatopancreas sample collection

Fifteen shrimp per tank (60 shrimp per experimental group) were anesthetized in ice water. Subsequently, the hepatopancreas was collected and rapidly frozen using liquid nitrogen. The hepatopancreas samples were stored at −80 °C until the analysis of gene expression.

### Quantitative real-time polymerase chain reaction (qRT-PCR)

To measure relative gene expression, total RNA was extracted from the shrimp hepatopancreas using TRIzol® (Sigma, MO, USA) following the manufacturer's instructions. The $OD_{260}/OD_{280}$ of the RNA extracts was measured using a Nano Drop 2000 (Thermo Scientific, NC, USA) to confirm the purity, which was between 1.8-2.0, indicating RNA extract. Afterwards, cDNA was synthesized using PrimeScript™ first-strand cDNA synthesis kit (Takara Bio, Shiga, Japan). PCR was conducted using the Thermal Cycler Dice™ Real Time System (TP950, TaKaRa Bio, Shiga, Japan), with a reaction mixture containing 2 µL of cDNA, 10 µL of TaKaRa Ex Taq™ SYBR premix, 0.4 µL of forward and reverse primers, respectively and 7.2 µL of $H_2O$ to make a total volume of 20 µL. The qRT-PCR analysis was performed using SYBR® Premix Ex Taq™ Perfect Real-Time Kits (RR820A, TaKaRa Bio, Shiga, Japan) with 1 cycle of 95 °C for 30 s and 40 cycles of 94 °C for 15 s, 58 °C for 20 s and 72 °C for 20 s. Relative expression levels of each gene were calculated based on the Pfaffl [24] method. *β-actin* was used as the housekeeping gene. qPCR efficiency (E, factor) was measured using a five-fold serial dilution of pooled cDNA, and efficiency (%) was calculated from the slope of the standard curve. Reference gene stability (M value) was evaluated using geNorm software [25] based on Ct values across all samples. *β-actin* showed an M value of 0.5, indicating suitable stability under the experimental conditions. The sequences of the primers used in this study are presented in Table 2.

### Ethical approval statements

Experimental protocols were conducted in accordance with the guidelines of the Animal Care and Use Committee of Jeju National University (2025−0073), South Korea. All procedures involving animals were approved by Jeju National University.

### Statistical analysis

For statistical analysis, percentage data were transformed using the arcsine square root method. Data were analyzed using a one-way analysis of variance (ANOVA). When significant differences among dietary groups were identified by ANOVA, Tukey's honestly significant difference post-hoc test was conducted for multiple comparisons between groups [26]. Results are expressed as mean ± standard deviation. All statistical analyses were performed using SPSS software (IBM SPSS Statistics 24, Corp., Armonk, NY, USA), with $P < 0.05$ considered statistically significant.

**Table 2. Primers used for real-time quantitative PCR.**

| Gene | Sequence (5′-3′) | Accession number | Efficiency (%) |
|---|---|---|---|
| *Lysozyme* | F-TGTTCCGATCTGATGTCC | AY170126.2 | 97.5 |
| | R-GCTGTTGTAAGCCACCC | | |
| *Crustin* | F-ACGAGGCAACCATGAAGG | AF430076 | 99.0 |
| | R-AACCACCACCAACACCTAC | | |
| *LGBP* | F-CAGGGGCAACGACAACTTTG | EU102286.1 | 96.3 |
| | R-GTGTGGGGATCTACTGCTCG | | |
| *Pen-3a* | F-CACCCTTCGTGAGACCTTTG | Y14926.1 | 98.2 |
| | R-AATATCCCTTTCCCACGTGAC | | |
| *proPO* | F-CGGTGACAAAGTTCCTCTTC | AY723296.1 | 97.8 |
| | R-GCAGGTCGCCGTAGTAAG | | |
| *FABP* | F: CGCTAAGCCCGTGCTGGAAGT | DQ398572.1 | 95.7 |
| | R: CTCCTCGCCGAGCTTGATGGT | | |
| *FAS* | F: CAGGTGGAGATGCTCCTCGTGTT | HM595630.1 | 96.9 |
| | R: GGTGACTAGCTCGGCTACATGGTT | | |
| *FATP* | F: GACGGGCAAAGCGACTGA | XM_027366808.1 | 98.5 |
| | R: ATGGACAAAGCCACGGAG | | |
| *CPT1* | F: ACTCCCGATAAGCACACC | XM_070121849.1 | 97.2 |
| | R: TTCATACATCCACCCCCT | | |
| *β-actin* | F: CCACGAGACCACCTACAAC | AF300705.2 | 100.2 |
| | R: AGCGAGGGCAGTGATTTC | | |

Abbreviations: *LGBP*, lipopolysaccharide-β-glucan binding protein; *Pen-3a*, penaeidin-3a; *proPO*, prophenoloxi-dase; *FABP*, fatty acid-binding protein; *FAS*, fatty acid synthase; *FATP*, fatty acid transport protein; *CPT1*, carnitine palmitoyl-transferase 1.

## Results

### Proximate composition and fatty acid profile of enriched rotifers and *Artemia*

In EXP-1, whole-body protein and crude lipid levels were significantly higher in all SCH-enriched rotifers than in RN and RCL rotifers (Table 3). Ash and moisture levels did not differ among groups. For whole-body fatty acids, myristic acid (C14) was not detected in RCL-enriched rotifers, and heptadecenoic acid (C17:1) appeared only in RCL-enriched rotifers. Palmitic acid (C16) and C22:6n3 levels were significantly higher in all SCH-enriched rotifers compared to RN and RCL rotifers. Stearic acid (C18) and oleic acid (C18:1n9c) levels were significantly higher in RN rotifers compared to all other enriched rotifers. Linoleic acid (C18:2n6c) and α-linolenic acid (C18:3n3) levels were significantly higher in RCL-enriched rotifers compared to all other enriched rotifers. In EXP-2, whole-body protein and crude lipid levels were significantly higher in all enriched *Artemia* than in AN *Artemia* (Table 4). Ash and moisture levels in all enriched *Artemia* did not show significant differences. Fatty acid analysis of the enriched *Artemia* showed that the levels of C16, palmitoleic acid (C16:1), C18, C18:1n9c, C18:2n6c, C18:3n3, C20:5n3 and C22:6n3 were significantly higher in all SCH-enriched *Artemia* compared to AN and ACL *Artemia*.

### Growth performance and survival

In EXP-1, FBL and LG of the shrimp were significantly increased in all SCH-enrichment groups compared to RN and RCL groups (Table 5). Survival was significantly higher in RS+T group than in RN and RCL groups. In EXP-2, FBW and WG of

**Table 3. Proximate composition and fatty acid profile of enriched rotifers.**

| | Rotifers | | | | |
|---|---|---|---|---|---|
| | RN | RCL | RS | RS+T | RS+I |
| *Proximate composition (%, wet basis)* | | | | | |
| Crude protein | 5.05±0.14[b] | 5.76±0.27[b] | 7.21±0.59[a] | 7.75±0.35[a] | 7.88±0.24[a] |
| Crude lipid | 9.35±0.31[b] | 10.5±0.36[b] | 13.4±0.57[a] | 13.4±0.15[a] | 13.9±0.12[a] |
| Crude ash | 0.86±0.05 | 0.87±0.04 | 0.82±0.05 | 0.84±0.05 | 0.86±0.04 |
| Moisture | 82.6±0.21 | 82.5±0.60 | 81.1±0.61 | 81.0±0.92 | 81.1±0.11 |
| *Fatty acid (%, rotifer)* | | | | | |
| C14 | 0.32±0.00[a] | 0.00±0.00[b] | 0.30±0.01[a] | 0.30±0.02[a] | 0.33±0.01[a] |
| C16 | 3.26±0.11[b] | 2.08±0.02[c] | 6.80±0.10[a] | 6.85±0.08[a] | 7.14±0.14[a] |
| C17:1 | 0.00±0.00[b] | 1.98±0.01[a] | 0.00±0.00[b] | 0.00±0.00[b] | 0.00±0.00[b] |
| C18 | 1.27±0.07[a] | 0.00±0.00[c] | 0.39±0.04[b] | 0.36±0.00[b] | 0.37±0.02[b] |
| C18:1n9c | 2.37±0.09[a] | 0.61±0.00[b] | 0.00±0.00[c] | 0.00±0.00[c] | 0.00±0.00[c] |
| C18:2n6c | 2.12±0.09[b] | 5.40±0.00[a] | 0.21±0.02[c] | 0.25±0.03[c] | 0.22±0.00[c] |
| C18:3n3 | 0.00±0.00[b] | 0.44±0.00[a] | 0.00±0.00[b] | 0.00±0.00[b] | 0.00±0.00[b] |
| C20:5n3 | ND | ND | ND | ND | ND |
| C22:6n3 | 0.00±0.00[b] | 0.00±0.00[b] | 5.65±0.10[a] | 5.65±0.12[a] | 5.86±0.11[a] |
| ∑Saturated fatty acids | 4.86±0.18[c] | 2.08±0.02[b] | 7.50±0.08[a] | 7.51±0.09[a] | 7.85±0.11[a] |
| ∑Monounsaturated fatty acids | 2.37±0.09[b] | 2.59±0.01[a] | 0.00±0.00[c] | 0.00±0.00[c] | 0.00±0.00[c] |
| ∑Polyunsaturated fatty acids | 2.12±0.09[b] | 5.84±0.01[a] | 5.86±0.08[a] | 5.90±0.09[a] | 6.08±0.11[a] |

The experimental groups included a non-enriched control (RN) and enriched groups (RCL, RS, RS+T and RS+I). Data are presented as mean±SD ($n=4$). Different superscripts in the same row indicate significant differences ($P<0.05$). ND, not detected.

**Table 4. Proximate composition and fatty acid profile of enriched *Artemia*.**

| | *Artemia* | | | | |
|---|---|---|---|---|---|
| | AN | ACL | AS | AS+T | AS+I |
| *Proximate composition (%, wet basis)* | | | | | |
| Crude protein | 5.86±0.02[b] | 6.06±0.04[a] | 6.04±0.04[a] | 6.16±0.00[a] | 6.07±0.02[a] |
| Crude lipid | 2.80±0.00[c] | 4.39±0.18[b] | 7.18±0.24[a] | 6.96±0.24[a] | 7.09±0.12[a] |
| Crude ash | 0.93±0.06 | 0.90±0.06 | 0.92±0.03 | 0.93±0.03 | 0.99±0.09 |
| Moisture | 86.1±0.52 | 86.5±0.69 | 86.9±1.10 | 86.8±0.54 | 86.4±0.55 |
| *Fatty acid (%, Artemia)* | | | | | |
| C16 | 0.41±0.02[c] | 0.64±0.00[b] | 1.40±0.00[a] | 1.31±0.04[a] | 1.39±0.05[a] |
| C16:1 | 0.13±0.00[c] | 0.19±0.02[b] | 0.27±0.01[a] | 0.28±0.00[a] | 0.29±0.00[a] |
| C18 | 0.32±0.00[c] | 0.48±0.00[b] | 0.67±0.01[a] | 0.68±0.01[a] | 0.68±0.01[a] |
| C18:1n9c | 0.70±0.00[c] | 1.08±0.03[b] | 1.42±0.04[a] | 1.40±0.02[a] | 1.39±0.00[a] |
| C18:2n6c | 0.21±0.00[c] | 0.40±0.01[b] | 0.46±0.01[a] | 0.45±0.00[a] | 0.45±0.00[a] |
| C18:3n3 | 0.94±0.00[d] | 1.46±0.02[c] | 1.93±0.01[a] | 1.86±0.01[b] | 1.89±0.02[ab] |
| C20:5n3 | 0.09±0.01[b] | 0.12±0.01[b] | 0.26±0.03[a] | 0.25±0.00[a] | 0.25±0.00[a] |
| C22:6n3 | 0.01±0.00[b] | 0.02±0.00[b] | 0.78±0.01[a] | 0.73±0.01[a] | 0.74±0.05[a] |
| ∑Sum saturated fatty acids | 0.73±0.01[c] | 1.12±0.00[b] | 2.07±0.01[a] | 2.00±0.03[a] | 2.07±0.04[a] |
| ∑Monounsaturated fatty acids | 0.83±0.00[c] | 1.27±0.01[b] | 1.69±0.03[a] | 1.68±0.01[a] | 1.68±0.00[a] |
| ∑Polyunsaturated fatty acids | 1.24±0.01[d] | 2.00±0.02[c] | 3.43±0.04[a] | 3.28±0.01[b] | 3.34±0.04[ab] |

The experimental groups included a non-enriched control (AN) and enriched groups (ACL, AS, AS+T and AS+I). Data are presented as mean±SD ($n=4$). Different superscripts in the same row indicate significant differences ($P<0.05$). ND, not detected.

**Table 5. Growth performance and survival of Pacific white shrimp *Penaeus vannamei* (initial body length: 0.67±0.03 mm) fed enriched rotifer for 9 days.**

| | Experimental groups | | | | |
|---|---|---|---|---|---|
| | RN | RCL | RS | RS+T | RS+I |
| Final body length[1] | 7.37±0.57[b] | 8.02±0.17[b] | 10.1±0.11[a] | 10.2±0.15[a] | 10.2±0.36[a] |
| Length gain[2] | 995±84[b] | 1091±25[b] | 1406±16[a] | 1413±23[a] | 1408±53[a] |
| Survival (%) | 43.5±11.7[c] | 55.1±9.3[bc] | 70.1±5.0[ab] | 73.9±6.5[a] | 68.5±3.3[ab] |

The experimental groups included a non-enriched control (RN) and enriched groups (RCL, RS, RS+T and RS+I). Data are presented as mean±SD (n=4). Different superscripts in the same row indicate significant differences ($P<0.05$).

[1]Final body length (mm).

[2]Length gain (%) = (Final body length – initial body length)/ (initial body length) × 100

shrimp were significantly increased in all SCH-enrichment groups compared to in AN and ACL groups (Table 6). Survival was not significantly different among all experimental groups.

## Fatty acid profile of whole-body of shrimp

In EXP-1, whole-body fatty acids of shrimp fed enriched rotifers for 9 days showed that all SCH groups had significantly higher C22:6n3 than RN and RCL (Table 7). RCL had higher C18:3n3, while RN showed significantly higher C16 and C18 than RS+T and RS+I groups. There were no significant differences in the C14, C16:1, C18:1n9c, C18:2n6c and C20:5n3 levels among all experimental groups. In EXP-2, whole-body fatty acids of shrimp fed enriched *Artemia* for 12 days showed that all SCH groups had significantly higher C22:6n3 than AN and ACL (Table 8). C16 levels were significantly higher in AN and ACL groups compared to all SCH enriched groups. C18 level was significantly higher in AN and ACL groups compared to AS group. C18:1n9c level was significantly higher in AN group compared to all other experimental groups. C18:3n3 level was significantly higher in ACL group compared to all other experimental groups. There were no significant differences in the C14, C16:1, C18:2n6c and C20:5n3 levels among all experimental groups.

## Relative gene expression related to lipid metabolism

In EXP-1, the shrimp hepatopancreas in all SCH-enriched groups showed significantly upregulated gene expressions of fatty acid-binding protein (*FABP*), fatty acid synthase (*FAS*), fatty acid transport protein (*FATP*) and carnitine palmitoyl-transferase 1 (*CPT1*) compared to RN and RCL groups (Fig 3). In addition, *FAS* gene expression was significantly upregulated in RCL group compared to RN group. In EXP-2, the shrimp hepatopancreas in all the SCH-enriched groups showed significantly upregulated gene expressions of *FABP*, *FAS*, *FATP* and *CPT1* compared to AN group (Fig 4).

**Table 6. Growth performance and survival of Pacific white shrimp *Penaeus vannamei* (initial body weight: 3.67±0.23 mg) fed enriched *Artemia* for 12 days.**

| | Experimental groups | | | | |
|---|---|---|---|---|---|
| | AN | ACL | AS | AS+T | AS+I |
| Final body weight[1] | 19.5±0.2[b] | 19.6±1.2[b] | 22.6±0.5[a] | 22.7±0.6[a] | 24.2±0.8[a] |
| Weight gain[2] | 430±6[b] | 434±33[b] | 515±14[a] | 518±16[a] | 559±23[a] |
| Survival (%) | 94.8±3.1 | 94.0±5.2 | 97.0±1.0 | 97.8±1.1 | 97.5±2.7 |

The experimental groups included a non-enriched control (AN) and enriched groups (ACL, AS, AS+T and AS+I). Data are presented as mean±SD (n=4). Different superscripts in the same row indicate significant differences ($P<0.05$).[1]Final body weight (mg)

[2]Weight gain (%) = [(final body weight - initial body weight)/ initial body weight] × 100

**Table 7. Fatty acid profile in whole body of Pacific white shrimp *Penaeus vannamei* fed the experimental rotifers for 9 days.**

| Fatty acid (%, lipids) | Experimental groups | | | | |
|---|---|---|---|---|---|
| | RN | RCL | RS | RS+T | RS+I |
| C14 | 4.66±0.00 | 4.53±0.28 | 4.41±0.02 | 4.49±0.18 | 4.34±0.17 |
| C16 | 29.5±0.40[a] | 21.8±1.96[ab] | 21.6±2.09[ab] | 20.5±2.91[b] | 21.4±0.88[b] |
| C16:1 | 4.26±0.15 | 4.12±0.13 | 4.08±0.07 | 4.19±0.24 | 4.12±0.14 |
| C18 | 26.3±0.01[a] | 19.1±0.22[b] | 19.1±0.29[b] | 19.4±0.52[b] | 19.3±0.07[b] |
| C18:1n9c | 4.78±0.25 | 4.43±0.15 | 4.32±0.00 | 4.41±0.56 | 4.22±0.15 |
| C18:2n6c | 7.49±0.24 | 7.55±0.43 | 7.56±0.34 | 7.39±0.25 | 7.46±0.12 |
| C18:3n3 | 5.68±0.30[b] | 18.3±0.04[a] | 5.69±0.28[b] | 5.55±0.10[b] | 5.59±0.53[b] |
| C20:5n3 | 8.56±0.10 | 11.4±1.48 | 11.1±0.29 | 11.1±0.40 | 11.1±0.75 |
| C22:6n3 | 8.76±0.29[b] | 8.80±0.23[b] | 22.2±1.50[a] | 23.0±0.64[a] | 22.5±0.53[a] |
| ∑Saturated fatty acids | 60.5±0.40[a] | 45.4±2.46[b] | 45.1±1.77[b] | 44.3±2.20[b] | 45.0±0.64[b] |
| ∑Monounsaturated fatty acids | 9.04±0.10 | 8.55±0.28 | 8.40±0.06 | 8.60±0.81 | 8.34±0.02 |
| ∑Polyunsaturated fatty acids | 30.5±0.32[b] | 46.1±2.18[a] | 46.6±1.84[a] | 47.1±1.39[a] | 46.6±0.63[a] |

The experimental groups included a non-enriched control (RN) and enriched groups (RCL, RS, RS+T and RS+I). Data are presented as mean±SD (n=4). Different superscripts in the same row indicate significant differences ($P < 0.05$).

**Table 8. Fatty acid profile in whole body of Pacific white shrimp *Penaeus vannamei* fed the experimental *Artemia* for 12 days.**

| Fatty acid (%, lipids) | Experimental groups | | | | |
|---|---|---|---|---|---|
| | AN | ACL | AS | AS+T | AS+I |
| C14 | 4.33±0.17 | 4.23±0.03 | 4.29±0.10 | 4.31±0.03 | 4.22±0.00 |
| C16 | 21.3±0.02[a] | 20.9±0.33[a] | 18.4±0.07[b] | 18.4±0.15[b] | 18.5±0.38[b] |
| C16:1 | 4.22±0.01 | 4.27±0.02 | 4.34±0.14 | 4.20±0.10 | 4.25±0.02 |
| C18 | 18.9±0.07[a] | 19.0±0.02[a] | 14.7±0.02[b] | 15.5±0.22[ab] | 16.3±2.17[ab] |
| C18:1n9c | 16.1±0.64[a] | 12.7±0.77[b] | 13.9±0.10[b] | 13.4±0.10[b] | 13.2±0.28[b] |
| C18:2n6c | 5.88±1.21 | 4.91±0.31 | 5.39±0.03 | 5.25±0.06 | 5.26±0.42 |
| C18:3n3 | 13.4±0.18[b] | 18.4±0.32[a] | 13.4±0.01[b] | 13.3±0.08[b] | 13.5±0.29[b] |
| C20:5n3 | 11.1±0.02 | 11.4±0.39 | 12.0±0.01 | 11.7±0.08 | 11.6±0.70 |
| C22:6n3 | 4.86±0.17[b] | 4.24±0.77[b] | 13.6±0.21[a] | 13.9±0.02[a] | 13.3±1.47[a] |
| ∑Saturated fatty acids | 44.5±0.25[a] | 44.1±0.34[a] | 37.4±0.20[b] | 38.3±0.04[b] | 39.0±1.79[b] |
| ∑Monounsaturated fatty acids | 20.3±0.63[a] | 16.9±0.75[b] | 18.2±0.04[b] | 17.6±0.00[b] | 17.4±0.26[b] |
| ∑Polyunsaturated fatty acids | 35.1±0.88[b] | 39.0±0.41[b] | 44.4±0.16[a] | 44.2±0.04[a] | 43.6±2.04[a] |

The experimental groups included a non-enriched control (AN) and enriched groups (ACL, AS, AS+T and AS+I). Data are presented as mean±SD (n=4). Different superscripts in the same row indicate significant differences ($P < 0.05$).

### Relative gene expression related to innate immunity

In EXP-1, *lysozyme* expression in the hepatopancreas of shrimp was significantly upregulated in all the enrichment groups compared to RN group (Fig 5). *Crustin*, lipopolysaccharide-β-glucan binding protein (*LGBP*) and prophenoloxidase (*proPO*) expression were significantly upregulated in all SCH enrichment groups compared to RN and RCL groups, with RS+I group showing the highest expression. Penaeidin-3a (*Pen-3a*) expression was significantly upregulated in all SCH enrichment groups compared to RN and RCL groups. In EXP-2, *lysozyme* expression in the hepatopancreas of shrimp was significantly upregulated in AS+T group compared to AN and ACL groups (Fig 6). *Crustin* expression was significantly

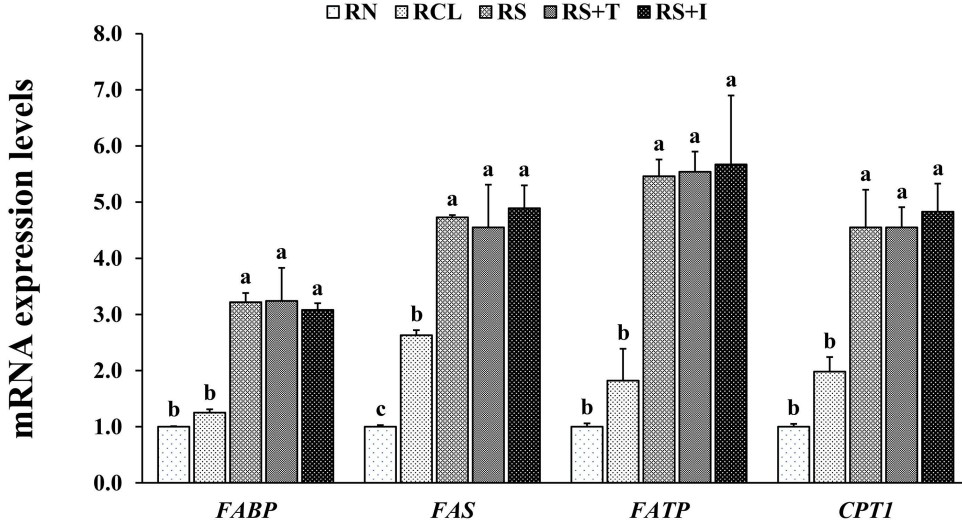

**Fig 3. Relative mRNA expressions of fatty acid-binding protein (*FABP*), fatty acid synthase (*FAS*), fatty acid transport protein (*FATP*) and carnitine palmitoyl-transferase 1 (*CPT1*) in the hepatopancreas of Pacific white shrimp *Penaeus vannamei* fed rotifers enriched with each treatment for 9 days.** Bars with different letters are significantly different ($P < 0.05$).

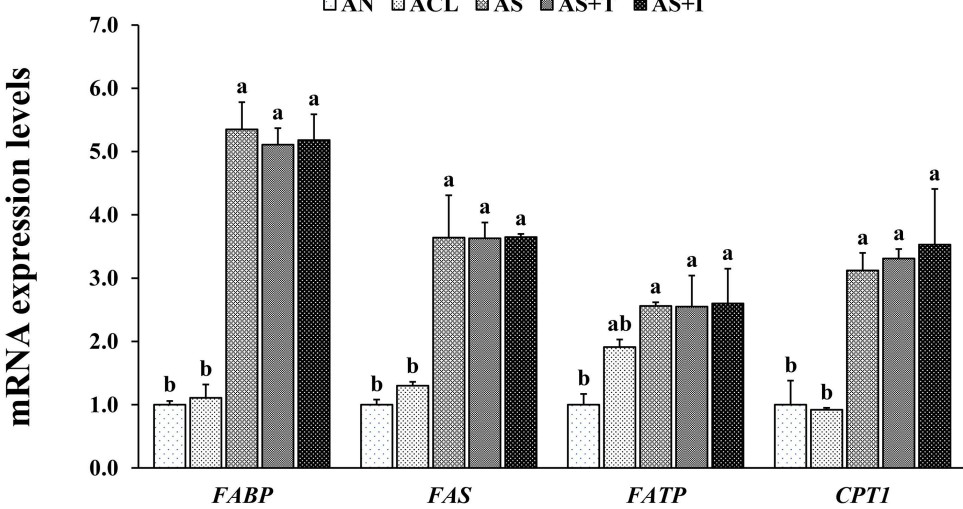

**Fig 4. Relative mRNA expressions of fatty acid-binding protein (*FABP*), fatty acid synthase (*FAS*), fatty acid transport protein (*FATP*) and carnitine palmitoyl-transferase 1 (*CPT1*) in the hepatopancreas of Pacific white shrimp *Penaeus vannamei* fed *Artemia* nauplii enriched with each treatment for 12 days.** Bars with different letters are significantly different ($P < 0.05$).

upregulated in all SCH enrichment groups compared to AN and ACL groups, with AS+I showed the highest expression. *LGBP* expression was significantly upregulated in all SCH enrichment groups compared to AN and ACL groups, with AS+T group which showed the highest expression. *Pen-3a* and *proPO* expression were significantly upregulated in AS, AS+T and AS+I groups compared to AN group.

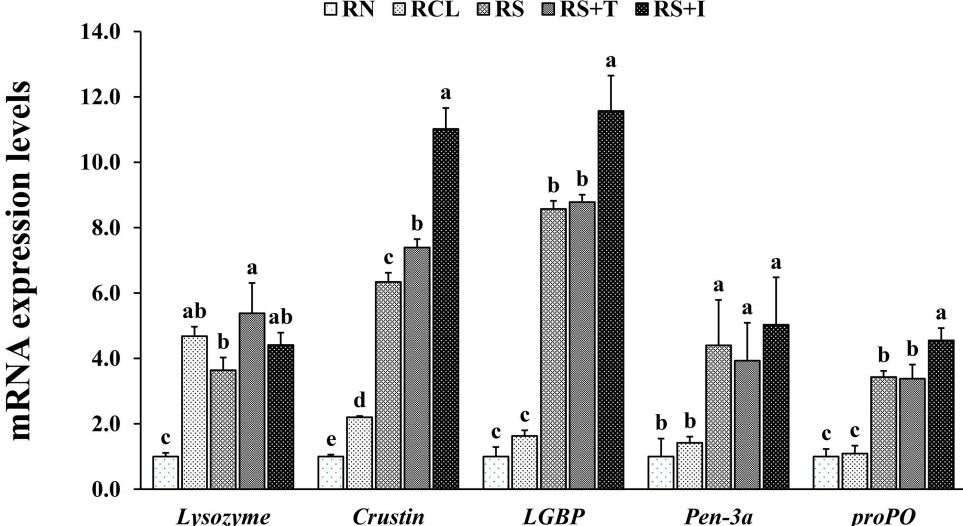

**Fig 5. Relative mRNA expressions of *lysozyme*, *crustin*, lipopolysaccharide-β-glucan binding protein (*LGBP*), penaeidin-3a (*Pen-3a*) and prophenoloxidase (*proPO*) in the hepatopancreas of Pacific white shrimp *Penaeus vannamei* fed rotifers enriched with each treatment for 9 days.** Bars with different letters are significantly different (*P*<0.05).

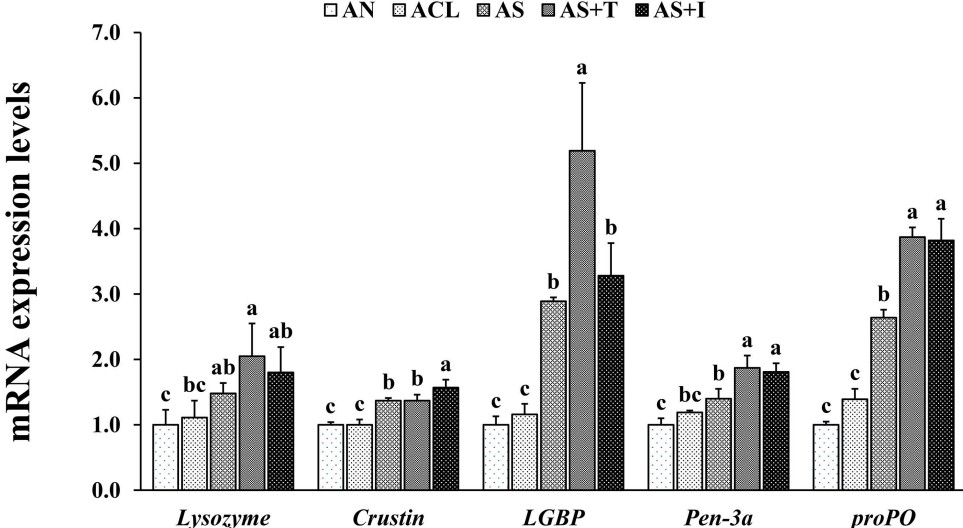

**Fig 6. Relative mRNA expressions of *lysozyme*, *crustin*, lipopolysaccharide-β-glucan binding protein (*LGBP*), penaeidin-3a (*Pen-3a*) and prophenoloxidase (*proPO*) in the hepatopancreas of Pacific white shrimp *Penaeus vannamei* fed *Artemia* nauplii enriched with each treatment for 12 days.** Bars with different letters are significantly different (*P*<0.05).

## Discussion

In the present study, the significant increase in DHA level in rotifers and *Artemia* enriched with SCH is consistent with previous studies showing that these live feeds can effectively serve as carriers of essential fatty acids such as DHA [5]. Similarly, previous studies on silver pompano and turbot have shown that feeding SCH-enriched rotifers or *Artemia* enhances

whole-body DHA content and improves both growth and survival [9,10]. SCH enrichment significantly improved growth in both feeding trials, with higher survival observed in EXP-1 compared to the non-enriched and *Chlorella*-enriched groups. These enhancements are likely linked to the increased dietary DHA supplied through SCH enrichment. Numerous studies have reported that increasing dietary DHA levels enhances the growth and survival of Pacific white shrimp [13,14,27]. Under high-salinity conditions (30 ppt), the shrimp's ability to synthesize polyunsaturated fatty acids (PUFA) such as EPA and DHA is limited, making dietary PUFA particularly important [14]. PUFA are required at high levels during larval stages due to their proportionately higher requirements in the development of visual and neural tissues as precursors for cell membrane structure, energy metabolism and hormone synthesis [28]. Suprayudi et al. [29] emphasized that a deficiency in dietary PUFA can lead to molting failure in crustacean larvae, which can significantly impact growth and survival. Overall, our findings suggest that dietary DHA supplementation through SCH enrichment of two species of live feed increases whole-body PUFA levels in shrimp, thereby improving their growth and survival.

The hepatopancreas is a vital organ responsible for lipid storage, synthesis, secretion of digestive enzymes and energy metabolism [30]. This study investigated the effects of dietary SCH on lipid metabolism in the hepatopancreas by examining the gene expression of *FAS*, *CPT1*, *FABP* and *FATP*. Shrimp fed SCH-enriched rotifers and *Artemia* showed higher expression levels of all lipid metabolism-related genes compared to those fed non-enriched or *Chlorella*-enriched live feeds. This upregulation corresponds to the increased PUFA (especially DHA), protein and total lipid levels in the enriched live feeds, which were transferred to the shrimp. Greater lipid availability provides more fatty acid substrates for synthesis and oxidation, while higher protein levels support the production of enzymes and regulatory proteins involved in lipid metabolism [31,32]. These nutritional improvements may activate lipid-sensitive transcription factors such as peroxisome proliferator-activated receptors and sterol regulatory element-binding proteins [28], thereby enhancing fatty acid-related gene transcription. Additionally, DHA incorporation into membrane phospholipids influences membrane properties and the function of membrane-bound proteins, including those involved in lipid metabolism [33]. These combined mechanisms explain how dietary SCH modulates lipid metabolism and supports shrimp health. Specifically, *FAS* catalyzes fatty acid synthesis and maintains lipid homeostasis [34]; *CPT1* regulates mitochondrial fatty acid oxidation for energy production [35]; *FABP* facilitates intracellular transport of fatty acids [36]; and *FATP* mediates fatty acid uptake across cell membranes, contributing to lipid utilization and energy balance [37]. In summary, our findings provide the first evidence that SCH-enriched live feeds enhance lipid metabolism in shrimp PL by improving dietary PUFA supply and proximate composition, thereby promoting growth, development, survival and energy metabolism.

This study highlights the potential of SCH as an immune-enhancing supplement in the PL culture of Pacific white shrimp. Enrichment of rotifers and *Artemia* with SCH upregulated the expression of key immune-related genes, including *lysozyme*, *crustin*, *LGBP*, *Pen-3a* and *proPO*. Given that invertebrates lack an adaptive immune system, the innate immune system is important in defending shrimp against pathogens [38]. *Lysozyme*, in particular, plays an important role in immune defense with its lytic activity against various cell wall pathogens, including *Vibrio* spp. [39]. *Crustin* and *Pen-3a* are antimicrobial peptides that provide protection against various pathogens, including Gram-positive bacteria and fungi [40]. *LGBP* is a pattern recognition protein that, upon binding to pathogen ligands activates the *proPO* system, which is crucial for melanization [38]. The enhanced immune gene expression observed in the SCH-enriched group may be largely explained by the increased PUFA levels in shrimp, as PUFA support immune regulation through roles in membrane structure, signaling and cellular function [28]. Chen et al. [41] reported that PUFA-driven induction of immune genes enhances antibacterial defense and improves resistance to *Vibrio parahaemolyticus*. SCH may further stimulate immunity through its polysaccharide components. Chen et al. [42] demonstrated that algal polysaccharides can activate hemocytes, promote *LGBP* binding, induce degranulation, and trigger the *proPO* cascade. Similarly, Chen et al. [43] found that Pacific white shrimp fed *Spirulina* powder showed increased resistance to *V. alginolyticus* due to activation of innate immune factors such as *lysozyme*, *LGBP* and phenoloxidase (PO). Collectively, these findings indicate that SCH functions as a potent immune stimulator in shrimp PL by enhancing immune gene expression and activating innate immune processes. Incorporating SCH into live feeds may therefore improve immunity and survival during the early stage of shrimp culture.

Our study demonstrated that the addition of taurine and IMP to the enrichment process yielded stronger immune-enhancing effects than SCH alone. Although research on the effects of taurine on innate immune gene expression in shrimp is limited, exogenous taurine has been reported to promote innate immune gene expression and enhance anti-bacterial activity in the Chinese mitten crab (*Eriocheir sinensis*) [44] and zebrafish (*Danio rerio*) [45]. Wang et al. [15] suggested that taurine metabolism is regulated in Pacific white shrimp infected with *Vibrio* spp., potentially enhancing antibacterial responses and survival. Additionally, dietary nucleotides, including IMP, are known to enhance immune responses in aquatic animals [18]. While the mechanisms underlying the immunostimulatory effects of nucleotides in shrimp remain unclear, Shankar et al. [46] demonstrated that dietary nucleotides stimulate innate immune responses, such as PO activity and total hemocyte count, in freshwater prawns (*Macrobrachium rosenbergii*), thereby improving resistance against *Aeromonas hydrophila*–induced white muscle disease. Similarly, Xiong et al. [47] found that feeding nucleotide-enriched yeast to Pacific white shrimp enhanced serum PO and lysozyme activities. Our findings are consistent with previous studies indicating that taurine and nucleotides can significantly enhance immune responses in various aquatic species. These findings align with our results, indicating that taurine and nucleotides substantially enhance immune responses across aquatic species. A synergistic interaction is likely, in which SCH improves PUFA availability while taurine and IMP modulate immune gene transcription and hemocyte activity. Although their immunostimulatory effects are evident, additional studies are needed to clarify the underlying mechanisms and determine optimal enrichment levels.

## Conclusion

Our study demonstrates that SCH enrichment of live feeds significantly increases DHA accumulation, leading to improved growth performance, lipid metabolism, innate immune responses, and survival in Pacific white shrimp PL. Moreover, co-supplementation with taurine and IMP showed further enhancement of immune gene expression. These findings highlight that the enrichment of live feeds with microalgal DHA sources and immunonutrients can be an effective nutritional strategy to improve the health and performance of shrimp during early developmental stages. Future research should focus on elucidating the molecular pathways involved in lipid utilization and immune modulation, as well as determining the optimal enrichment ratios for large-scale hatchery application.

## Supporting information

**S1 Fig. Measurement of total body length of Pacific white shrimp (*Penaeus vannamei*) post-larvae at the end of the feeding trials.** Photographs were taken to assess growth performance. No abnormal signs, such as shell discoloration, hepatopancreas color changes, melanization, body-wall damage, deformities, or abnormal pigmentation, were observed.
(DOCX)

**S2 Fig. Graphical abstract.**
(TIF)

## Author contributions

**Conceptualization:** Kyeong-Jun Lee.

**Formal analysis:** Sanghyun Song, Yeonji Lee, Yein Lee.

**Funding acquisition:** Young-Mi Lee, Kook-Jin Cho.

**Supervision:** Kyeong-Jun Lee.

**Writing – original draft:** Suhyeok Kim.

**Writing – review & editing:** Kyeong-Jun Lee.

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
