## [Decision Letter · Decision Letter 0]

7 Nov 2025

Dear Dr. Lee,

Thank you for submitting your manuscript to PLOS ONE. After careful consideration, we feel that it has merit but does not fully meet PLOS ONE’s publication criteria as it currently stands. Therefore, we invite you to submit a revised version of the manuscript that addresses the points raised during the review process.

We look forward to receiving your revised manuscript.

Kind regards,

Mohammed Fouad El Basuini, Professor

Academic Editor

PLOS ONE

Journal Requirements:

This work was supported by the CJ Cheiljedang Co., Ltd and Basic Science Research Program through the National Research Foundation of Korea (NRF) funded by the Ministry of Education (RS-2019-NR040078)

This work was supported by the CJ Cheiljedang Co., Ltd and Basic Science Research Program through the National Research Foundation of Korea (NRF) funded by the Ministry of Education (RS472 2019-NR040078)

This work was supported by the CJ Cheiljedang Co., Ltd and Basic Science Research Program through the National Research Foundation of Korea (NRF) funded by the Ministry of Education (RS-2019-NR040078)

6. Thank you for stating the following in the Competing Interests section:

The authors declare that they have no known competing financial interests or personal relationships that could have appeared to influence the work reported in this paper.

7. Thank you for stating the following in the Competing Interests/Financial Disclosure section:

The authors declare that they have no known competing financial interests or personal relationships that could have appeared to influence the work reported in this paper.

We note that one or more of the authors are employed by a commercial company: CJ Cheiljedang Co.

Additional Editor Comments:

If a reviewer suggests any modifications that conflict with the journal’s guidelines, please decline them. Ensure that all revisions strictly adhere to the journal’s requirements, and carefully proofread the manuscript before submission.

Reviewers' comments:

Reviewer's Responses to Questions

**Comments to the Author**

1. Is the manuscript technically sound, and do the data support the conclusions?

Reviewer #1: Yes

Reviewer #2: Yes

2. Has the statistical analysis been performed appropriately and rigorously?

Reviewer #1: Yes

Reviewer #2: Yes

3. Have the authors made all data underlying the findings in their manuscript fully available?

Reviewer #1: Yes

Reviewer #2: Yes

4. Is the manuscript presented in an intelligible fashion and written in standard English?

Reviewer #1: Yes

Reviewer #2: Yes

Reviewer #1: There are no recommendations/limitations

Conclusion less than enough

There are no ethical approval statements

There are no data availability

There is no declaration of conflict interests

Abstract

There are no highlights

There is no graphical abstract

What is /are the creativity of this work (ordinary study )

Divide the abstract into backgrounds/aims/methods/results and conclusion

LN/17—delete /short or running title ?

LN/20—more details are requested

LN/23---which types of nutrients are enriched

Huge number of abbreviations are detected—it better to create a separate table for this

Abstract is more than enough

LN/40-41—how ?

LN/43—add aquaculture industry/gene expression/growth performance /crustacean /immunity to the keywords

Introduction

LN/48/76/84—add references

LN/50—on culture water quality—clarify why/how

LN/53—microalgae—for ordinary readers more details are requested

LN/60—antioxidant capacity---mechanism?

Introduction is extremely long and repeated

Rewrite it again

Aims should be more clarified

Novelty should be more concentrated

Materials and Methods

LN/94-101—add reference and what about the physical and chemical nature of these materials

LN/104-108—delete

Table -1—should be directed toward the results

Figure(1&2)with the results

The whole experimental protocol has done according to whom

LN/176-178---2 different styles of writing references were detected—why ? same style should be

LN/186-189—why you did not do a histological section for more confirmation

LN/214—there is no reference for the statistical analysis

There is no plan for the study area

There is no IACUC-code –should be

Results

What about the clinical signs of the examined shrimp

What about the PM changes(gross figures are needed)

What about the mortality percent

LN/240---why did you put this table with results not at M&M

LN/241-247/250-256---be more summarized

8 tables and 6 figures—recheck more than enough

It is extremely very long

Rewrite it again

Discussion

It is very long –be more concise

Conclusions

Less than enough

References

Some cited references need to be more updated

Some cited references with missing data --recheck

Huge number of references were used–why

Some journal names were written abbreviated, while others were not—why ? same style should be

As volume/issue/pages/number—available—so no need for the link(s)—apply for all

Some cited references contained more than 6 authors—why—should be 6 at the maximum plus etal with the last ones---apply for all

All references should be rewritten

There are no gross figures—why

Reviewer #2: The manuscript presents a wonderful topic and was written perfectly. Many thanks to the authors, but some minor notes such as the abstract needs to explain many details of the materials and methods used and to explain some of the results digitally. the other notes are explained in the attached file.

**Do you want your identity to be public for this peer review?** For information about this choice, including consent withdrawal, please see our Privacy Policy

Reviewer #1: **Yes:** Elsayed Eldeeb Mehana Hamouda

Reviewer #2: No

---

## [Author Response · Author response to Decision Letter 1]

24 Nov 2025

Response Letter – ID: PONE-D-25-53314

Improvement of growth, lipid metabolism and innate immune response in Pacific white shrimp (Penaeus vannamei) post-larvae through enrichment of live feeds with Schizochytrium sp., taurine and inosine monophosphate

Dear Editor In-Chief,

Thank you for the thoughtful suggestions and comments on our manuscript. The authors have carefully revised the manuscript according to the comments/suggestions from reviewers and provided point-by-point response below. I anticipate your positive response and hope that you can find our manuscript suitable for the publication in PLOS ONE Journal.

Sincerely yours,

Kyeong-Jun Lee, Ph.D.

Responses to the comments of Editor

Answer: In accordance with the editor’s instructions, all files have been revised to fully comply with the PLOS ONE formatting and file-naming requirements.

Answer: The authority that approved access to the field site was Jeju National University, and this information has been included in the revised manuscript. The corresponding information has been included in the revised manuscript (Lines 468–471).

Answer: In accordance with the editor’s instruction, the Funding Information statement has been updated accordingly.

4. Thank you for stating the following financial disclosure: This work was supported by the CJ Cheiljedang Co., Ltd and Basic Science Research Program through the National Research Foundation of Korea (NRF) funded by the Ministry of Education (RS-2019-NR040078). Please state what role the funders took in the study. If the funders had no role, please state: "The funders had no role in study design, data collection and analysis, decision to publish, or preparation of the manuscript." If this statement is not correct you must amend it as needed. Please include this amended Role of Funder statement in your cover letter; we will change the online submission form on your behalf.

Answer: The funders contributed only by producing and supplying the Schizochytrium powder (CJ Cheiljedang Co., Ltd.) and by providing experimental tanks and facilities (National Research Foundation of Korea). The funders had no role in study design, data collection and analysis, decision to publish, or preparation of the manuscript.

5. Thank you for stating the following in the Acknowledgments Section of your manuscript: This work was supported by the CJ Cheiljedang Co., Ltd and Basic Science Research Program through the National Research Foundation of Korea (NRF) funded by the Ministry of Education (RS472 2019-NR040078). We note that you have provided funding information that is not currently declared in your Funding Statement. However, funding information should not appear in the Acknowledgments section or other areas of your manuscript. We will only publish funding information present in the Funding Statement section of the online submission form. Please remove any funding-related text from the manuscript and let us know how you would like to update your Funding Statement. Currently, your Funding Statement reads as follows: This work was supported by the CJ Cheiljedang Co., Ltd and Basic Science Research Program through the National Research Foundation of Korea (NRF) funded by the Ministry of Education (RS-2019-NR040078). Please include your amended statements within your cover letter; we will change the online submission form on your behalf. Please ensure that your manuscript meets PLOS ONE's style requirements, including those for file naming.

Answer: We mistakenly included funding information in the Acknowledgments section, although it should only appear in the Funding Statement. The funding-related text has now been removed from the Acknowledgments, and the correct Funding Statement has been included in the revised manuscript.

6. Thank you for stating the following in the Competing Interests section:

The authors declare that they have no known competing financial interests or personal relationships that could have appeared to influence the work reported in this paper. Please confirm that this does not alter your adherence to all PLOS ONE policies on sharing data and materials, by including the following statement: "This does not alter our adherence to PLOS ONE policies on sharing data and materials.” (as detailed online in our guide for authors http://journals.plos.org/plosone/s/competing-interests). If there are restrictions on sharing of data and/or materials, please state these. Please note that we cannot proceed with consideration of your article until this information has been declared. Please include your updated Competing Interests statement in your cover letter; we will change the online submission form on your behalf.

Answer: Following the editor’s instructions, we have revised the Competing Interests statement. The updated statement has been added to the revised manuscript.

7. Thank you for stating the following in the Competing Interests/Financial Disclosure section: The authors declare that they have no known competing financial interests or personal relationships that could have appeared to influence the work reported in this paper. We note that one or more of the authors are employed by a commercial company: CJ Cheiljedang Co. a. Please provide an amended Funding Statement declaring this commercial affiliation, as well as a statement regarding the Role of Funders in your study. If the funding organization did not play a role in the study design, data collection and analysis, decision to publish, or preparation of the manuscript and only provided financial support in the form of authors' salaries and/or research materials, please review your statements relating to the author contributions, and ensure you have specifically and accurately indicated the role(s) that these authors had in your study. You can update author roles in the Author Contributions section of the online submission form. Please also include the following statement within your amended Funding Statement. “The funder provided support in the form of salaries for authors [insert relevant initials], but did not have any additional role in the study design, data collection and analysis, decision to publish, or preparation of the manuscript. The specific roles of these authors are articulated in the ‘author contributions’ section.” If your commercial affiliation did play a role in your study, please state and explain this role within your updated Funding Statement. b. Please also provide an updated Competing Interests Statement declaring this commercial affiliation along with any other relevant declarations relating to employment, consultancy, patents, products in development, or marketed products, etc. Within your Competing Interests Statement, please confirm that this commercial affiliation does not alter your adherence to all PLOS ONE policies on sharing data and materials by including the following statement: "This does not alter our adherence to PLOS ONE policies on sharing data and materials.” (as detailed online in our guide for authors http://journals.plos.org/plosone/s/competing-interests) . If this adherence statement is not accurate and there are restrictions on sharing of data and/or materials, please state these. Please note that we cannot proceed with consideration of your article until this information has been declared. Please include both an updated Funding Statement and Competing Interests Statement in your cover letter. We will change the online submission form on your behalf.

Answer: Among the authors of this manuscript, Young-Mi Lee and Kook-Jin Cho are employed by the commercial company CJ Cheiljedang Co. They were involved in culturing and processing Schizochytrium to provide the Schizochytrium powder used as the experimental material. The specific roles of these authors are articulated in the “Author Contributions” section. We have updated both the Funding Statement and the Competing Interests Statement accordingly.

Answer: There were no such comments from the reviewers.

Responses to the comments of Reviewer #1.

1. Conclusion less than enough.

Answer: We appreciate the reviewer’s constructive suggestion. The conclusion section has been revised to provide a more comprehensive summary of the key findings, highlighting the synergistic effects of taurine and IMP supplementation, as well as the practical implications of Schizochytrium sp. enrichment for shrimp hatchery nutrition. The revised conclusion also outlines specific future research directions, including the molecular mechanisms and optimal enrichment levels. The updated text appears in the revised manuscript (Lines 441–450).

2. There are no ethical approval statements.

Answer: We have added a statement regarding ethical approval for animal experimentation, which was granted by Jeju National University. The corresponding information has been included in the revised manuscript (Lines 468–471).

3. There are no data availability.

Answer: We have included a data availability statement in the revised manuscript (Lines 473–474).

4. There is no declaration of conflict interests.

Answer: We have added a declaration of competing interests in the revised manuscript (Lines 476–479).

5. Abstract There are no highlights.

Answer: We have added a “Highlights” section to the revised manuscript (Lines 20–26).

6. There is no graphical abstract.

Answer: A graphical abstract has been prepared and will be submitted with the revised manuscript.

7. What is /are the creativity of this work (ordinary study).

Answer: This study provides novel insights into improving the growth, survival, and immune performance of Penaeus vannamei during the early post-larval stage, a critical developmental period characterized by high mortality rates. Nutritional enrichment of live feeds for shrimp post-larvae has been insufficiently investigated, despite its potential to enhance larval health and overall aquaculture productivity. Therefore, our work is innovative in that it demonstrates the practical and biological benefits of enriching zoo-live feeds with Schizochytrium sp., taurine and inosine monophosphate to improve early-stage shrimp performance. The novelty of this experiment has been described in the introduction (Line 94–100).

8. Divide the abstract into backgrounds/aims/methods/results and conclusion.

Answer: We thank the reviewer for the valuable suggestion. In response, the abstract has been revised to clearly present the background, aims, methods, results and conclusion in a logical flow.

9. LN/17—delete /short or running title ?

Answer: According to the journal’s submission guidelines, a short (running) title is required. Therefore, we have retained the running title and revised it to make it clearer and more concise while maintaining the journal’s format (Line 28–29).

10. LN/20—more details are requested.

Answer: We appreciate the reviewer’s comment. The sentence has been revised to provide more detailed information about the parameters evaluated in this study (Lines 32–35). Specifically, we have clarified that growth performance, lipid metabolism, immune response and survival of Penaeus vannamei post-larvae were assessed.

11. LN/23---which types of nutrients are enriched.

Answer: The term “nutrients” in this sentence refers to Schizochytrium sp., taurine, and inosine monophosphate (IMP), which were used to enrich the live feeds. In contrast, no enrichment additives were applied to the live feeds in the control group. To avoid possible confusion, the term “nutrients” has been replaced with “unenriched rotifers and unenriched Artemia” in the revised manuscript (Lines 37, 39).

12. Huge number of abbreviations are detected—it better to create a separate table for this.

Answer: We agree with the reviewer’s comment. Accordingly, the abbreviations used in the abstract and main text have been clearly defined in the caption of Table 2.

13. LN/40-41—how ?

Answer: The statement was based on the observed results in this study. Shrimp fed with nutritionally enriched live feeds showed improved growth performance, survival and lipid metabolism. In addition, immune-related genes were upregulated in groups supplemented with taurine and IMP in combination with Schizochytrium sp. Therefore, the sentence was written to reflect these findings.

14. LN/43—add aquaculture industry/gene expression/growth performance /crustacean /immunity to the keywords.

Answer: As suggested by the reviewer, the keywords “aquaculture industry”, “gene expression”, “growth performance”, “crustacean” and “immunity” have been added to the revised manuscript (Line 53–54).

15. LN/48/76/84—add references.

Answer: We thank the reviewer for the valuable comment. A reference has been added to support the statement (Line 58). The content (Line 84–87) was based on the findings of previous studies [7–14], which were cited to substantiate the expected effects discussed in this section. Similarly, the statement (Line 94–95) was supported by references [15–18], which provide the scientific basis for the study’s objectives and predicted outcomes.

16. LN/50—on culture water quality—clarify why/how.

Answer: We thank the reviewer for the valuable comment. In the early post-larval stages, the low mobility of larval often results in uneaten micro- or particulate feeds settling at the bottom, which can deteriorate water quality. Numerous studies have reported that live feeds cause less water pollution than formulated diets because they remain suspended longer and are more readily consumed by larvae (Bengtson, 2003; Gonçalves et al., 2024; Melaku et al., 2024).

Bengtson, D.A., 2003. Status of marine aquaculture in relation to live prey: past, present and future. Live feeds in marine aquaculture. pp.1-16.

Goncalves, R., Pfalzgraff, T., & Lund, I. (2024). Impact of live feed substitution with formulated diets on the development, digestive capacity, biochemical composition, and rearing water quality of European lobster (Homarus gammarus, L.) larvae. Aquaculture, 586, 740776.

Melaku, S., Geremew, A., Getahun, A., Mengestou, S., & Belay, A. (2024). Challenges and prospects of using live feed substitutes for larval fish. Fisheries and Aquatic Sciences, 27(8), 475-487.

17. LN/53—microalgae—for ordinary readers more details are requested.

Answer: We thank the reviewer for the suggestion. The sentence has been revised to provide more detail for general readers, explaining that microalgae are rich in essential fatty acids, particularly DHA and EPA, and serve as effective nutritional sources for enriching live feeds to enhance growth, lipid metabolism and immunity in aquatic animal larvae. The revised sentence appears in the manuscript at Lines 64–67.

18. LN/60—antioxidant capacity---mechanism?

Answer: Carotenoids such as astaxanthin and β-carotene, as well as vitamins including niacin and pyridoxine, are well-known to enhance antioxidant capacity in shrimp (Zhang et al., 2013; Liu et al., 2024). These bioactive compounds, abundant in Schizochytrium sp., can increase the activities of antioxidant enzymes (e.g., superoxide dismutase, catalase, and glutathione peroxidase), thereby improving overall antioxidant defenses when incorporated into feeds or used to enrich live feeds (Dineshbabu et al., 2019).

Dineshbabu, G., Goswami, G., Kumar, R., Sinha, A., Das, D., 2019. Microalgae–nutritious, sustainable

---

## [Decision Letter · Decision Letter 1]

15 Jan 2026

Improvement of growth, lipid metabolism and innate immune response in Pacific white shrimp (Penaeus vannamei) post-larvae through enrichment of live feeds with Schizochytrium sp., taurine and inosine monophosphate

PONE-D-25-53314R1

Dear Dr. Lee,

We’re pleased to inform you that your manuscript has been judged scientifically suitable for publication and will be formally accepted for publication once it meets all outstanding technical requirements.

Kind regards,

Mohammed Fouad El Basuini, Professor

Academic Editor

PLOS One

Additional Editor Comments (optional):

Reviewers' comments:

Reviewer's Responses to Questions

**Comments to the Author**

Reviewer #1: All comments have been addressed

Reviewer #3: All comments have been addressed

2. Is the manuscript technically sound, and do the data support the conclusions?

Reviewer #1: Yes

Reviewer #3: Yes

3. Has the statistical analysis been performed appropriately and rigorously?

Reviewer #1: Yes

Reviewer #3: Yes

4. Have the authors made all data underlying the findings in their manuscript fully available?

Reviewer #1: Yes

Reviewer #3: Yes

5. Is the manuscript presented in an intelligible fashion and written in standard English?

Reviewer #1: Yes

Reviewer #3: Yes

Reviewer #1: Accepted after all corrections have been done

The paper is fall in the scope and aims of the journal

Reviewer #3: I have carefully reviewed the revised manuscript. The authors have satisfactorily addressed all of the previous comments and concerns. The manuscript is now suitable for publication in its current form.

**Do you want your identity to be public for this peer review?** For information about this choice, including consent withdrawal, please see our Privacy Policy

Reviewer #1: **Yes:** Elsayed Eldeeb Mehana

Reviewer #3: No

---

## [Editor Report · Acceptance letter]

PONE-D-25-53314R1

PLOS One

Dear Dr. Lee,

I'm pleased to inform you that your manuscript has been deemed suitable for publication in PLOS One. Congratulations! Your manuscript is now being handed over to our production team.

Kind regards,

on behalf of

Prof. Mohammed Fouad El Basuini

Academic Editor

PLOS One